

# Targeted correction of megabase-scale *CNTN6* duplication in induced pluripotent stem cells and impacts on gene expression

Maria Gridina[1,2,3], Polina Orlova[1,3] and Oleg Serov[1]

[1] Genomic Mechanisms of Ontogenesis, Institute of Cytology and Genetics, Novosibirsk, Novosibirsk, Russia
[2] Ontogenetics, Research Institute of Medical Genetics, Tomsk National Research Medical Center of the Russian Academy of Sciences, Tomsk, Russia
[3] Natural Sciences, Novosibirsk State University, Novosibirsk, Russia

## ABSTRACT

Copy number variations of the human *CNTN6* gene, resulting from megabase-scale microdeletions or microduplications in the 3p26.3 region, are frequently implicated in neurodevelopmental disorders such as intellectual disability and developmental delay. However, duplication of the full-length human *CNTN6* gene presents with variable penetrance, resulting in phenotypes that range from neurodevelopmental disorders to no visible pathologies, even within the same family. Previously, we obtained a set of induced pluripotent stem cell lines derived from a patient with a *CNTN6* gene duplication and from two healthy donors. Our findings demonstrated that *CNTN6* expression in neurons carrying the duplication was significantly reduced. Additionally, the expression from the *CNTN6* duplicated allele was markedly lower compared to the wild-type allele. Here, we first introduce a system for correcting megabase-scale duplications in induced pluripotent stem cells and secondly analyze the impact of this correction on *CNTN6* gene expression. We showed that the deletion of one copy of the *CNTN6* duplication did not affect the expression levels of the remaining allele in the neuronal cells.

## INTRODUCTION

The peritelomeric region pter-p26.3 of human chromosome 3 encompasses three genes from the immunoglobulin superfamily: *CHL1*, which encodes the neural cell adhesion molecule L1-like protein, and *CNTN4* and *CNTN6*, which encode contactin-4 and contactin-6 proteins, respectively. Copy number variations (CNVs) in this region are associated with severe neurodevelopmental disorders including intellectual disability (ID), developmental delay, autism spectrum disorders (ASD), seizures, and attention deficit hyperactivity disorder (*Shoukier et al., 2013*; *Kashevarova et al., 2014*; *Moghadasi et al., 2014*; *Te Weehi et al., 2014*; *Hu et al., 2015*; *Murdoch et al., 2015*; *Palumbo et al., 2015*; *Li et al., 2016*; *Huang et al., 2017*; *Mercati et al., 2017*; *Tassano et al., 2018*). Among 14 patients with deletions (seven cases) or duplications (five cases) involving only the *CNTN6*

Corresponding author
Maria Gridina, gridinam@gmail.com

gene or its fragments, eleven exhibited neurodevelopmental disorders, and six displayed dysmorphic features (*Hu et al., 2015*). Notably, two unrelated patients presented with both neurodevelopmental and neuropsychiatric disorders, as well as dysmorphic features—one with a microdeletion and the other with a microduplication solely involving the *CNTN6* gene. Similar observations were noted in three families with ID, psychomotor retardation, microcephaly, and dysmorphic features, each carrying a microdeletion or microduplication of just the *CNTN6* gene, and in two boys with ID and ASD, one with a microdeletion and the other with a microduplication of the *CNTN6* gene, both of paternal origin (*Kashevarova et al., 2014*; *Tassano et al., 2018*). These cases highlight a notable overlap in neuropsychiatric and dysmorphic features among patients with either deletions or duplications of the *CNTN6* gene. Another common aspect of *CNTN6* gene deletions and duplications is haploinsufficiency, albeit with incomplete penetrance and no known cases of homozygous duplication carriers. In contrast, despite the severe neurodevelopmental issues associated with CNVs of the *CNTN6* gene, deleterious point mutations (such as missense, nonsense, splice site, or disrupted start or stop codons) in this gene typically do not exhibit clinical phenotypes and were identified in only about 1% of a cohort of 942 healthy individuals (*Murdoch et al., 2015*).

The observed similarities in neurodevelopmental disorders among patients with microdeletions and microduplications of the *CNTN6* gene were clarified in our previous experiments using induced pluripotent stem (iPS) cells (iPS cells). These cells were derived by reprogramming skin fibroblasts from healthy donors and a patient with *CNTN6* gene duplication, followed by their *in vitro* differentiation into neurons (*Gridina et al., 2018*). The duplication spans 943,993 bp on chromosome 3 (Chr3:560 685–1 504 678, human reference genome GRCh37/hg19). It encapsulates the entire *CNTN6* gene, extending approximately 600 kb upstream of the gene's 5′ end and 50 kb downstream of its 3′ end. The duplication is arranged in a tandem "head-to-tail" fashion without additional rearrangements within the *CNTN6* gene sequences. The directed neural differentiation was induced either by the exogenous expression of the neurogenin 2 (Ngn2) transcription factor or through spontaneous neural differentiation *via* the neural rosette pathway. Our analysis of *CNTN6* expression in these neuronal cells showed that the level of expression was significantly reduced compared to the neuronal cells from healthy donors (*Gridina et al., 2018*). Moreover, we observed that gene expression in cells with the duplication was reduced by more than twofold compared to normal cells. This reduction can be explained by the possibility that the *CNTN6* gene is expressed monoallelically in neurons, while the duplication significantly decreases *CNTN6* expression from the second allele. It is important to note that no additional abnormalities were found in the *CNTN6* gene or its regulatory regions.We assume that the duplication of *CNTN6* leads to the development of neurodevelopmental disorders through a decrease in the gene expression. It is consistent to suppose that correction of this CNV can lead to correction of the phenotype (*Maino et al., 2021*; *Lemoine et al., 2024*).

Genome editing technology enables precise manipulation of DNA sequences, allowing stable modifications through targeted knockouts, insertions, and replacements. The CRISPR/Cas9 system is the most widely employed genome editing technology due to its

high editing efficiency, ease of use and cost-effectiveness (*Doudna & Charpentier, 2014*; *Knott & Doudna, 2018*). Site-specific recombinases such as Cre-loxP are also highly effective tools for genome modification (*Skarnes et al., 2011*; *Flemr & Bühler, 2015*). Both Cre-loxP and CRISPR-Cas9 are widely used for genome editing and are powerful tools, which are extensively used in various biological studies, including screenings in cell cultures and the creation of genome-modified animal models. Cre-loxP is best suited for the insertion, deletion, or rearrangement of large DNA fragments, such as hundreds or thousands of base pairs. On the other hand, CRISPR/Cas9 excels in precise, targeted modifications like point mutations and gene knockouts; however, challenges remain with the deletion of large fragments (*Zhang et al., 2024*).

Combining Cre/loxP with CRISPR/Cas9 technology can accelerate conditional gene targeting and enhance genome-editing efficiency (*Yang et al., 2017*; *Noiman & Kahana, 2018*; *Thakur & Welford, 2020*). In the present work, we also combined Cre/loxP with CRISPR/Cas9 technology for targeted correction of the megabase-scale duplications and to evaluate the impact of this correction on the expression of the remaining copies.

## MATERIALS AND METHODS

### Human iPS cells culture conditions and electroporation

Human iPS cells were previously generated by reprogramming skin fibroblasts (*Gridina et al., 2018*). iPS cells were cultured and expanded under feeder-free conditions, using TeSR-E8 medium (Stemcell Technologies, Vancouver, Canada) in dishes coated with Matrigel (BD Biosciences, Franklin Lakes, NJ, USA), according to the manufacturer's recommendations.

Electroporation was utilized to insert loxP sites at the targeted locations within the second copy of the *CNTN6* gene in human iPS cells. The cells were cultured in TeSR-E8 medium, supplemented with 1% Penicillin-Streptomycin (Pen-Strep) on Matrigel-coated dishes (according to the manufacturer's manual). The iPS cells were seeded in a 6-well plate ($30 \times 10^3$ cells per cm$^2$). Electroporation was performed when 70% confluency was reached. One hour before electroporation, the medium was switched to TeSR-E8 without antibiotics, but with the addition of 10 μM ROCK inhibitor. The cells were treated with TrypLE solution, rinsed with DMEM/F12 medium containing 10% KnockOut Serum Replacement, and then washed with phosphate buffered saline (PBS). After removing the supernatant thoroughly, the cells were resuspended in Buffer R (Neon™ electroporation system Kit) at a concentration of $180 \times 10^3$ cells/μl. Transfection was carried out using the Neon™ electroporation system in a 10 μl volume under the following conditions: 1,200 V and 30 μs per pulse. The iPS cells were transfected with a combination of 2 μM single-stranded oligodeoxynucleotides (ssODN) and 0.5 μg of the pSpCas9(BB)-2A-Puro (PX459) V2.0 plasmid (#62988; Addgene, Watertown, MA, USA) or 1 μg of the Cre-IRES-PuroR plasmid (#30205; Addgene, Watertown, MA, USA). Guide RNA and ssODN sequences were designed using the online resource Benchling (https://www.benchling.com). Post-electroporation, the cells were cultured on Matrigel in TeSR-E8 medium with ROCK inhibitor and without antibiotics. Twenty-four hours after electroporation, 0.4 μg/ml puromycin was added to the culture medium for selection, which was conducted

over the next 3 days. Ten days later, emerging clones were manually transferred to 24-well plates.

## Droplet digital polymerase chain reaction

Droplet digital polymerase chain reaction (ddPCR) was conducted using a Bio-Rad QX100 system, adhering to the manufacturer's guidelines. The reaction volume was 20 µl, consisting of 1× ddPCR™ Supermix for Probes (no dUTP), 900 nM primers, 250 nM probes, and the DNA template. The sequences for primers and probes are provided in Table S1. Each sample's ddPCR reactions were performed in duplicate. Droplets were generated using DG8™ cartridges and transferred to PCR plates, then analyzed using a droplet reader with Quanta Soft software (Bio-Rad, Hercules, CA, USA). The procedure was replicated for three independent biological replicates. The human *ACTB* gene was used as the reference gene. The absolute number of *CNTN6* complementary DNA (cDNA) copies obtained *via* ddPCR was normalized to the cDNA copies of the *ACTB* gene.

## Statistical analysis

All statistically analyzed data followed a normal distribution, as confirmed by the Shapiro-Wilk test. The statistical significance was then assessed using a Student's t-test.

## Ngn2-induced differentiation of iPS cells to iN cells

Induced neuronal (iN) cells were generated from iPS cells *via* the forced expression of the neurogenin-2 transcription factor (encoded by the Ngn2 gene), as previously described (Gridina et al., 2018). In brief, we utilized lentiviral constructs: FUW-TRE Ngn2/Puro, containing full-length mouse Ngn2 and puromycin-resistance genes, and FUW-TRE EGFP, containing Green Fluorescent Protein (GFP). Lentiviruses were produced in the Phoenix cell line using Lipofectamine 3000 (Invitrogen, Waltham, MA, USA), following the manufacturer's recommendations. The iPS cells were transduced with these lentiviruses in a culture medium supplemented with 5 µg/ml polybrene (MilliporeSigma, Burlington, MA, USA). To initiate Ngn2 gene expression, 2 µg/ml doxycycline (DOX; Sigma-Aldrich, St. Louis, MI, USA) was added to the medium 1 day post-transduction, which consisted of DMEM/F12, 1% MEM NEAA, 1% Pen-Strep, 1% N2 supplement, human brain derived neurotrophic factor (BDNF) (10 ng/ml), mouse laminin (0.2 µg/ml; Sigma-Aldrich, St. Louis, MI, USA), and human Neurotrophin-3 (NT-3) (10 ng/ml; PeproTech, Cranbury, NJ, USA). The Ngn2-expressing cells were selected using 1 µg/ml puromycin (Gibco, Waltham, MA, USA) starting the day after DOX induction. Following 1 day of puromycin selection, the cells (45 × 103 cells/cm2) were plated on mitomycin C-treated mouse glial cells. The cultures were then maintained for 3 weeks in Neurobasal Medium supplemented with 2% B27, 1% GlutaMAX™-I, 1% Pen-Strep, human BDNF (10 ng/ml), and NT-3 (10 ng/ml, all from Gibco, Waltham, MA, USA). Total RNA was subsequently extracted from the iN cells for droplet digital PCR analysis.

## RESULTS AND DISCUSSION

### Insertion of loxP sites flanking a single copy of the duplicated *CNTN6* gene in iPS cells

To remove one copy of the duplicated *CNTN6* gene in iPS cells, we utilized a combination of CRISPR/Cas9-mediated genome editing and the Cre/loxP system. The experimental design for the deletion is outlined in Fig. 1A. The initial step involves inserting loxP sites at the boundaries of the duplicated segments. Guide RNA (gRNA) target sites were selected 300 bp before the end of the duplicated region (Table S1). Consequently, there are three target breaks in the genome of iPS cells carrying the duplication. The first is at the end of the first duplicated copy (marked as "1"), the second is associated with the second duplication copy ("2"), and the third is on the wild-type *CNTN6* allele ("3"). A single-stranded oligodeoxynucleotide (ssODN) was engineered to include homology arms, a loxP site, and a HindIII restriction enzyme site (Table S1). The iPS cells iTAFdup22 were obtained from the skin fibroblast of patients with *CNTN6* duplication (*Gridina et al., 2018*). We electroporated iTAFdup22 cells by the pSpCas9(BB)-2A-Puro plasmid (#62988, Addgene, Watertown, MA, USA), which contains the gRNA, Cas9 gene, and puromycin resistance gene, along with the ssODN. As a result, we obtained 68 clones, designated iTAFdup22-L1-1 through iTAFdup22-L1-68.

We genotyped iPS cell clones using HindIII digestion of PCR products obtained with two sets of primers (Table S1): "1" for detecting loxP site insertion between two duplicated copies and "2,3" for detecting loxP site insertion at the end of the duplicated region, as well as in the wild-type allele (Fig. 1A). The analysis revealed that the obtained clones could be categorized as follows (Table 1):

1) 35 clones without any insertions;
2) Clones where the double-strand break repair in region #1 likely occurred *via* non-homologous end joining (NHEJ), resulting in the loss of the junction between duplicated copies (Fig. 1B, clone 67), while having loxP site insertions in both regions #2 and #3;
3) Clones where double-strand break repair likely occurred *via* NHEJ in regions #1, #2, and #3 (Fig. 1B, clones 50, 61);
4) Clones without an insertion in region #1, but with loxP site insertions in both regions #2 and #3 (Fig. 1B, clone 27);
5) Clones without an insertion in region #1, but with loxP site insertions in either region #2 or #3 (Fig. 1B, clone 48);
6) Clones with a loxP site insertion in region #1 and either region #2 or #3 (Fig. 1B, clone 66);
7) Nine clones with loxP site insertions in all three regions (Fig. 1B, clone 6).

The correct insertion of full-length loxP sites was confirmed by Sanger sequencing (Fig. 1C). The selected clone was subsequently subcloned to minimize potential heterogeneity within the cell population.

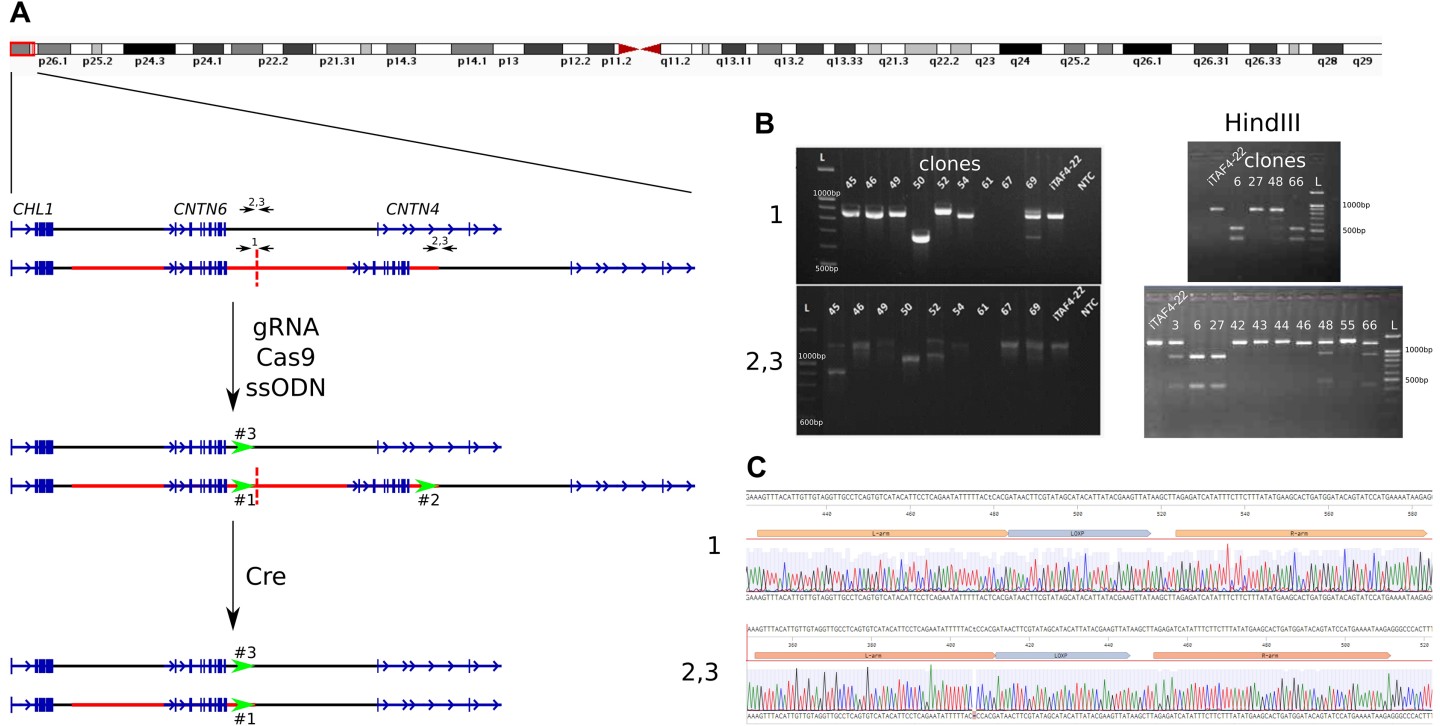

**Figure 1 Insertion of loxP sites flanking a duplicated copy.** (A) Schematic overview of the 3p26.3 region in normal and duplicated alleles, alongside the experimental setup. The duplicated region is marked by a red line, with copies separated by a dashed line. Black arrows indicate primer pairs used for detecting insertions, while green arrows represent the positions of loxP sites. (B) Example of genotyping of iPS cells clones by PCR and HindIII digestion to detect loxP site insertions. Displayed are PCR products from chromosome 3 region using primer pairs "1" (detect the end of the first copy), and "2,3" (detect the end of duplicated region and the same time the wild-tipe allele). Clones 67 and 69 show a loss of genetic material at the junction of two duplicated copies (region #1), clones 45 and 52 show deletions at the end of the duplication (regions #2 and #3), and clone #50 exhibits deletions in both regions simultaneously. Clones 6, 8, 15, 16, 24, 31, 32 show incretion of loxP and HindIII sites. iTAF4-22 is parent iPS cells before genome editing, NTC–the non-template control, and L–the 100 bp DNA length marker. (C) Sequence analysis confirming the insertion of loxP sites at positions "1" (the end of the first copy) and "2, 3" (the end of duplicated region and the same time the wild-tipe allele).

## Excising a region flanked by loxP sites

A subclone was chosen for further manipulation, and the cells were transfected with the Cre-IRES-PuroR plasmid (#30205; Addgene), which contains the Cre recombinase. As a result, 11 subclones were obtained, labeled iTAFdup-cre1 through iTAFdup-cre11. Analysis of these clones revealed that two lacked PCR products corresponding to the junction of the duplicated copies (Figs. 2A, 2B).

To verify the correction of chromosomal rearrangements, we analyzed the ratio of parental alleles of the *CNTN6* gene within the duplicated region. The sixth exon of the *CNTN6* gene contains single-nucleotide polymorphisms (SNPs) that discriminate the parental alleles (*Gridina et al., 2018*). DdPCR was conducted on DNA samples isolated from iTAFdup22-cre2 and iTAFdup-cre3, as well as iTAFdup22 (parental iPS cells with the duplication), and iTAF1nor36 (iPS cells derived from healthy donor fibroblasts lacking the duplication). The results demonstrated that after correcting the chromosomal rearrangements, the ratio of parental alleles was nearly 1:1, 0.987 and 0.998 in iTAFdup-cre2 and iTAFdup-cre3 clones, respectively. This ratio matched that of iTAF1nor36 cells

**Table 1 Genotyping iPS cell clones.**

| Number of clones | Integration site | | | Illustration on Fig. 1B |
|---|---|---|---|---|
| | "1" | "2" | "3" | |
| 35 | – | – | – | |
| 6 | NHEJ | + | + | Clone 67 |
| 4 | NHEJ | NHEJ | NHEJ | Clones 50, 61 |
| 8 | – | + | + | Clone 27 |
| 4 | – | "2" or "3" | "2" or "3" | Clone 48 |
| 2 | + | "2" or "3" | "2" or "3" | Clone 66 |
| 9 | + | + | + | Clone 6 |

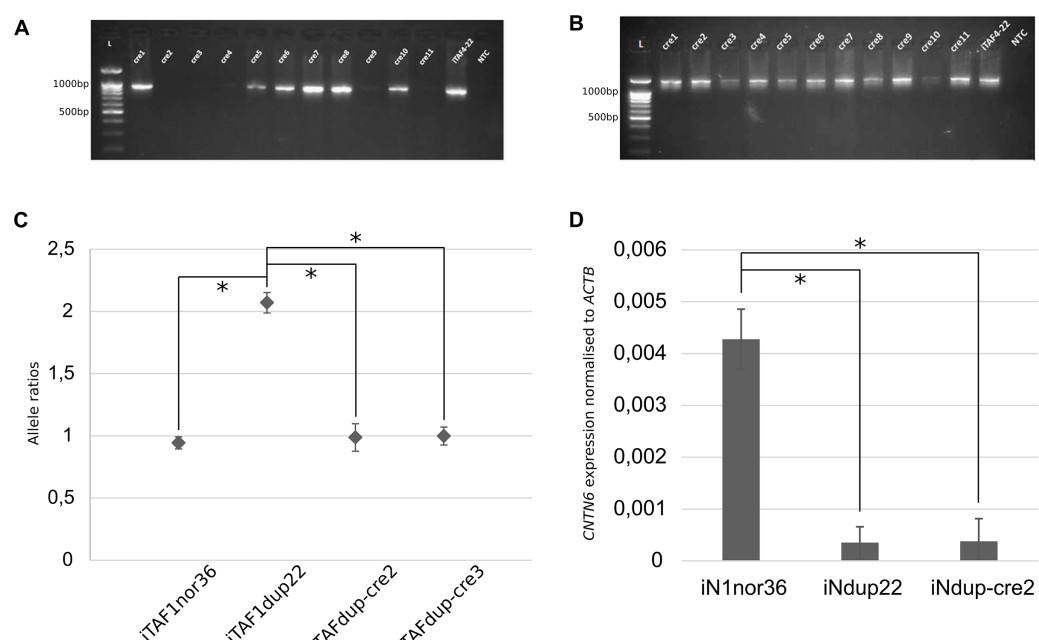

**Figure 2 Correction of duplication in iPS cells.** (A) Genotyping of iPS cell clones by PCR to detect the junction of two duplicated copies. Displayed are PCR products from chromosome 3 region using primer pair "1". iTAF4-22 represents parent iPS cells before genome editing, NTC is the non-template control, and L–the 100 bp DNA length marker. (B) PCR products from chromosome 3 region using primer pairs "2,3". iTAF4-22 indicates parent iPS cells, NTC is the non-template control, and L–the 100 bp DNA length marker. (C) Ratios of *CNTN6* alleles estimated by ddPCR for iPS cells from a healthy donor (iTAF1nor36), a patient with a *CNTN6* duplication (iTAFdup22), and iPS cells with corrected duplication (iTAFdup-cre2, iTAFdup-cre3). The error bar estimate 95% confidence interval of the measurement (*$p < 0.01$, Student's t-test). (D) Comparison of *CNTN6* transcript levels in iN cells from a healthy donor (iN1nor36), a patient with the duplicated *CNTN6* gene (iNdup22), and after duplication correction (iNdup-cre2) after 3 weeks of neural differentiation. *CNTN6* gene expression was normalized to the human *ACTB* gene. Data are presented as mean ± SD (*$p < 0.05$, Student's t-test).

from a healthy donor. Conversely, parental iTAFdup22 cells with duplication exhibited an allele ratio of 2:1 (Fig. 2C). The fact that exon 6 of the *CNTN6* gene is located at a distance of 167 kb from the introduced CRISPR/Cas9 break, allows us to be confident that the

obtained results are not a consequence of damage to the primer landing site. Thus, the chromosomal correction successfully normalized the copy number of the *CNTN6* gene in the corrected cell lines.

### *CNTN6* expression in Ngn2-induced neurons

Previously we demonstrated that *CNTN6* expression in neurons carrying the duplication was significantly reduced compared with expression in neurons from healthy donors. Moreover, the expression from the exactly duplicated allele was markedly lower compared to the wild-type allele (*Gridina et al., 2018*). To assess impacts of duplication correction on *CNTN6* gene expression levels, we examined iN cells derived from three different sources: healthy donor iPS cells (iN1nor36), patient iPS cells with a duplicated *CNTN6* gene (iNdup22), and iPS cells where one copy of the duplicated *CNTN6* gene was removed (iNdup-cre2). These cells underwent neuronal differentiation induced by the forced expression of exogenous Ngn2. The results of the quantitative analysis of *CNTN6* gene expression across these clones, based on three replicates, are presented in Fig. 2D. We confirmed previous findings that the *CNTN6* expression level in iN cells from patients with a gene duplication was lower than that in iN cells from a healthy donor. However, the removal of one duplicated copy in iNdup-cre2 cells did not increase the expression level of the remaining *CNTN6* gene copy. This rather disconcerting result may indicate that this duplication may lead to the establishment of some epigenetic marks in early development that are not erased during reprogramming into iPS cells. "Epigenetic memory" can be retained in iPS cells, even though these cells meet all commonly accepted criteria for pluripotency (*Lin, Lehle & McCarrey, 2024*).

## CONCLUSION

We present a straightforward and efficient method for correcting megabase-scale duplications in human iPS cells. This approach combines two powerful platforms: CRISPR-Cas9 and Cre-loxP systems. The correction process is achieved in two simple steps. First, targeted integration of loxP sites is performed at the borders of the rearrangement in iPS cells. In the second step, Cre recombinase is introduced *via* a vector, enabling precise excision of the duplicated copy. Given that we had previously demonstrated that gene duplication results in decreased *CNTN6* gene expression, though the mechanism was unclear, we anticipated that correcting this rearrangement would reverse the effect. However, even after the precise removal of one of the duplicated copies, *CNTN6* gene expression remained lower than in control cells and was comparable to the levels observed in the original cells before correction.

In summary, the findings suggest that the transcriptional status, whether active or repressive, of both copies in a duplication remains unchanged in the remaining copy after one is removed.

### Funding

The study was supported by the Russian Science Foundation No-21-65-00017. The funders had no role in study design, data collection and analysis, decision to publish, or preparation of the manuscript.

### Grant Disclosures

The following grant information was disclosed by the authors:
Russian Science Foundation: 21-65-00017.

### Competing Interests

The authors declare that they have no competing interests.

### Author Contributions

- Maria Gridina conceived and designed the experiments, performed the experiments, analyzed the data, prepared figures and/or tables, authored or reviewed drafts of the article, and approved the final draft.
- Polina Orlova conceived and designed the experiments, performed the experiments, analyzed the data, prepared figures and/or tables, authored or reviewed drafts of the article, and approved the final draft.
- Oleg Serov conceived and designed the experiments, analyzed the data, authored or reviewed drafts of the article, and approved the final draft.

### Data Availability

The raw data is available in the Supplemental File. Cell culture was performed at the Collective Center of ICG SB RAS "Collection of Pluripotent Human and Mammalian Cell Cultures for Biological and Biomedical Research", project number FWNR-2022-0019 (https://ckp.icgen.ru/cells/; http://www.biores.cytogen.ru/brc_cells/collections/ICG_SB_RAS_CELL).

### Supplemental Information

Supplemental information for this article can be found online at http://dx.doi.org/10.7717/peerj.18567#supplemental-information.

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
