# Peer review of "Targeted correction of megabase-scale CNTN6 duplication in induced pluripotent stem cells and impacts on gene expression"

_PeerJ, doi:10.7717/peerj.18567_

## Round 0.1 · original submission · Minor Revisions

The two reviewers have acknowledged the value of this paper, but they have pointed out some issues that need to be revised. I think their comments are appropriate for improving this paper. Please revise the paper according to the reviewers' comments. I look forward to receiving your revised manuscript.

Reviewer 1 ·

Basic reporting

The manuscript by Gridina and colleagues presents a novel genetic tool for correcting megabase-scale duplications in induced pluripotent stem cells (iPSCs). The authors first validate the proof-of-concept for this newly developed tool and then examine its impact on CNTN6 gene expression. Their results show that the removal of one duplicated copy in iNdup-cre2 cells does not increase the expression level of the remaining CNTN6 gene copy. This suggests that the duplication may establish epigenetic marks early in development, which are not erased during reprogramming into iPSCs. This represents an important contribution to understanding the role of CNTN6 gene duplication in neurodevelopmental disorders.
The authors employ clear, professional language, and the manuscript is well-structured with high-quality figures and sufficient background information. However, the following changes are recommended:
Line 76: 'It is consistent to suppose that correction of this CNV can lead to correction of the phenotype.' Adding a few references in the introduction section to support this statement would provide readers with more information about the other tools currently available for targeted correction of megabase-scale duplications in the literature.
Lines 38 to 73 present various pieces of evidence supporting the conclusion that CNTN6 duplication contributes to neurodevelopmental disorders. However, the primary focus of this study is to develop a method for the targeted correction of megabase-scale duplications. Therefore, a more detailed discussion of the analytical tools available for detecting and correcting such duplications should be included in the introduction section.

Experimental design

The manuscript presents original research that aligns with the journal's scope, addressing the development of genetic tools for correcting megabase duplications in the CNTN6 gene. The study includes thorough investigations; however, the methods section requires additional details to ensure clarity and reproducibility.
For example:
a) Line 86-87: 'The cells were cultured in TeSR-E8 medium supplemented with 1% Pen-Strep on Matrigel-coated dishes.' Please specify the number of cells cultured, the volume of the dishes, and the amount of Matrigel used for coating the dishes.
b) Line 91: Please provide the composition of Buffer R, or if it is a commercial product, include the specific vendor.
c) Please include a brief paragraph at the end of the Methods section describing the statistical analysis, detailing how it was conducted.

Validity of the findings

I commend the experimental conclusions, which are well-stated and clearly linked to the original research question. However, one suggestion is that incorporating a Student’s t-test in Figure 2C would help demonstrate the robustness of the experiment.

Additional comments

The manuscript is well-written and addresses an important research question. I recommend its acceptance with minor revisions.

Reviewer 2 ·

Basic reporting

The main finding of this article is that correcting the megabase-scale duplication of the human CNTN6 gene in patient-derived iPSCs did not affect the expression level of CNTN6. To achieve these results, the authors created iPSCs with loxP sites flanking the megabase-scale duplication, allowing for the removal of the duplication using Cre recombinase. The method employed by the authors is very clear; however, it appears that the authors conducted some experiments using only one clone, despite having generated multiple clones. Therefore, I support its publication in PeerJ after some revisions.

Experimental design

The authors did not specify which subclone was chosen for the Cre recombinase transduction experiment. If the selected clone had any abnormalities, the results may have been influenced by the specific characteristics of that clone.

Validity of the findings

1) From line 165, the authors explain Fig. 1B and Table 1, but it is difficult to understand how the clones are categorized. Please clarify how the clones are categorized based on the PCR results. Additionally, please include a representative gel image for each category.

2) To draw a conclusion about the expression level of CNTN6 in Fig. 2D, the authors need to analyze at least each two different clones derived from two distinct parental clones out of the nine shown in Fig. 1.

Additional comments

Interestingly, the expression level of CNTN6 in iTAFdup22 is less than half that of iN1nor36, which seems to extend to the expression of non-overlapping alleles in these patient iPS cells. Can the authors address this in the discussion section?

---

## Round 0.2 · accepted · Accept

I confirmed that the authors have addressed all of the reviewer's comments.